# Participation in Sports Clubs during College Is an Important Factor Associated with School Counselors’ Participation in Leisure Time Activities

**DOI:** 10.3390/ijerph19095632

**Published:** 2022-05-05

**Authors:** Pei-Fung Wu, Ya-Ping Ke

**Affiliations:** 1Department of Kinesiology, Health and Leisure Studies, National University of Kaohsiung, Kaohsiung 811726, Taiwan; yapin0113@qtm.kh.edu.tw; 2Kaohsiung Municipal Ciao Tou Junior High School, Kaohsiung 82543, Taiwan

**Keywords:** IPAQ, physical activity participation, attitude toward physical activity, self-perceived health

## Abstract

This study explored the physical activity participation, barriers to physical activity, attitudes toward physical activity, and physical activity levels of full-time school counselors at junior high schools in Kaohsiung City, Taiwan. The survey was conducted by means of a questionnaire. A total of 156 questionnaires were distributed, 137 were returned, and 130 were valid. The reliability and constructed validity of the questionnaire were analyzed using Cronbach’s alpha coefficient and factor analysis, respectively. The participants’ International Physical Activity Questionnaire (IPAQ) scores were estimated to assess their physical activity levels. Simple and multiple regression analyses were performed to analyze the effects of independent variables on the respondents’ physical activity levels and attitude toward physical activity. Of the respondents surveyed, 44.6%, 36.9%, and 18.5% had low, moderate, and high levels of physical activity according to their IPAQ scores, respectively. In total, 55.4% of the school counselors met the World Health Organization criteria for physical activity. Moreover, the results of the Kruskal–Wallis test showed that respondents with high levels of physical activity, aged ≥41, and who perceived themselves to be healthy were more likely to have participated in sports clubs during their college years. Furthermore, participation in sports clubs during college years and self-perceived health were significant predictors of attitude toward physical activity. In conclusion, participation in sports clubs during college was an important factor related to school counselors’ physical activity.

## 1. Introduction

Physical inactivity has been identified as the fourth leading risk factor for mortality worldwide [1]. The current trend in physical inactivity is partly attributable to insufficient participation in physical activity during leisure time and an increase in sedentary behavior during occupational and domestic activities [2]. Physical activity is one of the most crucial factors affecting individuals’ physical health. In the era of modern technology, people are increasingly prioritizing convenience, and many lead sedentary lifestyles [3]. As a result, they are more inclined to be diagnosed with chronic diseases such as hypertension, cardiovascular diseases, obesity, and atherosclerosis [4,5]. The impact on children is even greater; therefore, physical fitness and health of children and adolescents are of concern worldwide [6,7,8]. Ferrari et al. (2015) concluded that the physical fitness of schoolchildren has gradually declined over the past 30 years [9]. Thus, understanding the factors that make people physically active or not can help in developing strategies to promote physical activity.

In Taiwan, since 2012, the Ministry of Education has established a policy on the establishment of school counselors in primary and secondary schools. In particular, schools in junior high schools are required to employ at least one full-time school counselor, and thereafter, additional full-time school counselors will be employed each year according to the total number of classes in the school. In addition, in 2014, the 12-year national education curriculum entered full implementation. In the face of changing education policies and reforms, secondary school teachers are being asked to devote more time and effort to teaching and counseling. These education policy reforms have also further increased the heavy workloads of teachers. School counsellors, who are mainly responsible for individual and group counseling of students, have more sitting time than lecturing teachers. Therefore, the ability of school counsellors to maintain sufficient physical activity to focus on their work and achieve optimal counseling outcomes deserves further study.

Exercise is a part of physical activity that is programmatic, structural, and repetitive and has the intermediary or ultimate goal of balancing energy and improving or maintaining physical health [10]. Regarding body structure and composition, exercise can enhance the density and support of bone tissue, correct body posture, and help individuals control their weight and maintain their physique [11]. On the other hand, exercise can also increase muscle mass, burns surplus calories, enhances the flexibility of joints, increases blood flow to the heart, and shortens the time required for the body to return to normal after strenuous activity [12]. In terms of disease prevention and control, exercise can prevent and control hypertension and reduce the incidence of noninsulin-dependent diabetes [13]. It can also help alleviate lower back pain and prevent cancer [14]. Studies have also shown that exercise can help people relax, improve their sleep quality, and cope with stress, thereby improving their quality of life [15]. Furthermore, at the level of mental and spiritual activities, exercise can boost self-esteem, promote learning, stabilize emotions, enhance creativity, improve self-confidence, and reduce anxiety [16].

World Health Organization (WHO) guidelines recommend that adults aged 18–64 years should do at least 150–300 min of moderate-intensity aerobic physical activity, at least 75–150 min of vigorous-intensity aerobic physical activity, or an equivalent combination of moderate- and vigorous-intensity activity every week [17]. Data from the 2017 survey by the Sports Administration of Taiwan’s Ministry of Education, reveal that 33.2% of the respondents fulfilled the “7333” recommendation of regular exercise (exercise at least three times a week for 30 min performed at an intensity that causes a heart rate of at least 130 or sweating and panting) [18]. Of the respondents, the group comprising participants ≥60 years old contained the highest proportion of people who fulfilled the 7333 recommendation (61.8%), followed by those comprising participants 13–17 years old (43.5%), 18–29 years old (30.9%), 50–59 years old (20.9%), 30–39 years old (20.5%), and 40–49 years old (20.2%). When the respondents were categorized by occupation, 51.5% of retired/nonworking people; 40.1% of students; 39.5% of stay-at-home mothers; 36% of military personnel, civil servants, and teachers; 24.5% of professional specialists; and 19.4% of laborers met the 7333 recommendations. However, 14.7% of the respondents self-reported no exercise. The highest proportion of this population was 30–39 years old (19.4%), followed by 40–49 years old (18.5%). In the occupational classification, the military, civil servants, and teachers are usually grouped together, but in reality these three occupations have completely different job attributes and should not be placed in the same category, especially when conducting physical activity surveys.

Several studies have explored the barriers responsible for physical inactivity [19,20,21]; however, exploring the factors affecting rates of regular exercise among particular occupational populations is worthwhile. In addition to understanding the factors for physical inactivity and eliminating barriers, it is crucial to explore the factors promoting physical activity so that different strategies can be developed to encourage physical activity at different stages. In Taiwan, the position of school counselor has been established since 2012. Although the establishment of secondary school counselors is that of teachers, they do not teach classes. School counselors are responsible for the second and third levels of the school’s three-level preventive counseling model. As a result, they have more opportunities to sit and interview or counsel students, and they spend more time sitting than other teachers. While the physical activity levels of schoolteachers have been explored [22,23,24], the physical activity levels of school counselors, who are also teachers but have a very different job nature, have not been explored. Therefore, the aim of this study was to assess the physical activity of school counselors. Our results demonstrated that 55.4% of the school counselors met the WHO criteria for physical activity. Our results also demonstrated that the participation in sports clubs during college is a key factor for school counselors’ physical activity.

## 2. Materials and Methods

The questionnaire employed in this study comprised a demographic characteristics survey, the short-form Taiwan version of the IPAQ [25], and components assessing the respondents’ physical activity participation, perceived barriers to physical activity, and attitudes toward physical activity. The use of the short-form Taiwan version of the IPAQ was approved by Taiwan’s Ministry of Health and Welfare. The seven items of IPAQ identified the total minutes spent on moderate- and vigorous-intensity physical activity, walking, and inactivity over the seven days prior to survey administration.

### 2.1. Respondents

In December 2010, the administrative district of Kaohsiung County was merged into Kaohsiung City. The junior high school counselors in this study were recruited from schools in the administrative district of Kaohsiung City before the merger. All the junior high school counselors (*n* = 156 from 89 schools) employed in Kaohsiung City during the first semester of 2018 were invited to complete the questionnaire. The responses to the questionnaire were anonymous, non-interactive, and non-interventionist, and no specific individuals could be identified from the information in the questionnaire. According to Cohen’s (1988) criterion, the sample size for the regression model was calculated to be 119, from 10 predictors, with an alpha value of 0.05, a power of 0.80, and a medium effect size by using G* power software [26,27].

### 2.2. Questionnaire Design

The physical activity participation component of the questionnaire was designed to assess the physical activity participation of the full-time school counselors before and after their employment as full-time school counselors to determine whether they had maintained their physical activity habits and to determine whether exercise habits and participation in sports clubs during college affected the counselors’ habits later in life. The physical activity barrier component was modified from Dergance et al. (2003), which was designed to determine the factors that prevented the study participants from participating in physical activity, including lack of peers, inadequate facilities, lack of support, poor physical strength, heavy workload, and lack of interest [28]. The physical activity attitude questionnaire was adapted from the Leisure Attitude Scale developed by Ragheb and Beard (1982) to evaluate the participants’ attitudes toward physical activity [29]. Each item on the questionnaire was scored on a 5-point Likert-type scale, with scores ranging from 5 to 1 for “strongly agree”, “agree”, “neither agree nor disagree”, “disagree”, and “strongly disagree”; higher scores indicated higher degrees of participation, greater perception of barriers, and more positive attitudes on each of the respective components [30].

### 2.3. Scoring of the IPAQ

The energy expenditure for each activity level is represented as a metabolic equivalent (MET) score: walking = 3.3, moderate activity = 4.0, vigorous activity = 8.0 [31]. Each participant’s physical activity level was measured in MET minutes per week (MET-min/week); MET minutes per week were calculated by multiplying the MET value of an activity by the minutes for which the activity was performed and by the number of days in a week on which the activity was undertaken (MET × minutes of activity × days per week). Any activity lasting less than 10 min was not counted. According to the scoring protocol, the participants’ physical activity levels were categorized as low (0 ≤ MET-min/week < 600), moderate (600 ≤ MET-min/week < 3000), or high (MET-min/week ≥ 3000) [31,32].

### 2.4. Statistical Analysis

The demographic characteristics were analyzed using descriptive statistics. The reliability analysis was performed using the Cronbach’s alpha coefficient. Construct validity was evaluated using factor analysis. Following analysis through the Kaiser–Meyer–Olkin test and Bartlett’s spherical test (KMO > 0.7; *p* < 0.05), exploratory factor analysis was conducted for each item of the questionnaire’s components. The correlation matrix for the questions was analyzed through principal component extraction, the maximum variance method, ranking according to factor loadings, and evaluation of coefficients with hidden absolute values less than 0.4. Nonparametric analysis was used when the data were not assumed to come from a prescribed model. The Spearman correlation analysis was performed to analyze the correlation between variables. Multiple regression analysis was used to predict the independent variables of attitude toward physical activity. In multiple regression model, the Shapiro–Wilk test, Durbin–Watson test, Cook’s distance, and variance inflation factor (VIF) were used to examine the normality, the correlation of residual terms, the outlier, and the collinearity, respectively. The Scheffe post hoc test or multiple comparisons of Kruskal–Wallis test was used to investigate the association within variables. Spearman correlation analysis and regression analysis were performed using SigmaPlot software, version 13.0 (Systat Software, San Jose, CA, USA). Descriptive statistics analysis, reliability analysis, exploratory factor analysis, calculation of the standardized beta coefficient, Scheffe post hoc test, Kruskal–Wallis test, Durbin–Watson test, and Cook’s distance of multiple regression were performed using SPSS software, version 22.0 (IBM, Armonk, NY, USA). The significance levels of the study were set at *p* < 0.05.

## 3. Results

The questionnaire surveys were distributed from September 2018 to October 2018. Table 1 lists the items of the constructed questionnaire obtained after factor analysis. After the dimensionality reduction factor analysis, Cronbach’s alpha values for the physical activity participation, barrier, and attitude components of the questionnaire changed 0.866, 0.64, and 0.875 to 0.857, 0.71, and 0.902, respectively. Moreover, the Cronbach’s alpha values for component 1 (participation in physical activity after employed) and component 2 (participation in sports clubs during college years) of physical activity participation were 0.84 and 0.841, respectively. In total, 156 full-time school counselors from 89 junior high schools in Kaohsiung were invited to complete the questionnaires, and 137 questionnaires were collected, which reflected a response rate of 87.82%. After exclusion of seven invalid questionnaires, 130 valid questionnaires were analyzed, reflecting a validity rate of 83.33%.

### 3.1. Demographic Characteristics of School Counselors

Of the 130 respondents, 104 (80%) were female and 26 (20%) were male. Regarding age, 60 (46.2%) of the respondents were 31–40 years old, 44 (33.8%) were 21–30 years old, 26 (20%) were over 41 years old. Details of the respondents’ height, body weight, marital status, education, and self-perceived health status are shown in Table 2.

### 3.2. Physical Activity Levels of School Counselors

As shown in Table 3, among these respondents, 44.6% had low levels of physical activity (<600 MET-min/week), 36.9% had moderate levels of physical activity (≥600 MET-min/week <3000), and 18.5% had high levels of physical activity (MET-min/week ≥3000). The data revealed that 55.4% of school counselors met the physical activity criteria of WHO guidelines. The IPAQ scores of low, moderate, and vigorous groups were 260.9 ± 177.3, 1302.7 ± 603.8, and 6909.9 ± 3700.7, respectively. The accumulated MET-min/week for walking and vigorous activity was 526.6 ± 462.9 and 518.3 ± 529.5, respectively, in the moderate physical activity group, and 2593.3 ± 2190.7 and 2540.0 ± 2509.2, respectively, in the high physical activity group; this finding indicates that walking and vigorous activity may be the most common physical activities among school counselors. It is worth noting that the scores for physical activity had a multiplicative factor additive effect. The multiplicative coefficients for walking, moderate activity, and vigorous activity were 3.3, 4.0, and 8.0, respectively. Therefore, the more intense the activity, the higher the score obtained, especially for vigorous activity.

### 3.3. Variables Associated with Physical Activity Levels

As indicated in Table 4, the respondents’ participation in sports clubs during college years (rho = 0.234, *p = 0*.007), attitude toward physical activity (rho = 0.365, *p = 0*.000), age (rho = −0.179, *p* = 0.042), body weight (rho = 0.221, *p* = 0.012), and self-perceived health status (rho = 0.186, *p* = 0.034) were significantly associated with their physical activity levels. Physical activity levels (rho = 0.237, *p* = 0.007), attitude toward physical activity (rho = 0.439, *p = 0*.000), age (rho = −0.341, *p* = 0.000) and self-perceived health status (rho = 0.219, *p* = 0.012) were significantly associated with participation in sports clubs during college years. Moreover, the results of the Kruskal–Wallis test showed that respondents with high levels of physical activity (χ^2^ = 7.843, *p* = 0.02), age ≥41 years (χ^2^ = 17.302, *p* = 0.000), and self-perceived as healthy (χ^2^ = 10.434, *p* = 0.005) had more participation in sports clubs during their time at college. Furthermore, self-perceived health status (rho = 0.257, *p* = 0.003) was significantly associated with attitude toward physical activity. Interestingly, age had a significant negative correlation with physical activity level (rho = −0.179, *p* = 0.042) and participation in sports clubs during college years (rho = −0.341, *p* = 0.000).

### 3.4. Determinants Associated with Attitude toward Physical Activity

As indicated in Table 5, the respondents’ participation in sports clubs during college years (β = 0.409, t = 5.106, *p* = 0.000, VIF = 1.051) and self-perception as being healthy (β = 0.168, t = 2.096; *p* = 0.038, VIF = 1.051) were significant determinants associated with attitude toward physical activity. In the diagnosis of the regression model, the Durbin–Watson value of 2.320 was within the acceptable range (1.5~2.5) indicating that the residues were relatively independent. The Cook’s distance was less than 1 (0.000~0.108) for all data points indicating no outliers. The VIF showed a value less than 10.0 indicating lack of collinearity between independent variables. The Shapiro–Wilk test with a *p* = 0.375 indicates the variable was normally distributed. Moreover, the F test of the regression model was significant (F = 18.472, *p* = 0.000). With an alpha value of 0.05, the power was 1.0 (>0.8), indicating that the participation in sports clubs during college years, as well as how healthy school counselors perceived themselves to be, was an important factor in determining attitude toward physical activity.

## 4. Discussion

The aim of this study was to assess the physical activity of school counselors who had more opportunities to sit and interview or counsel students. Our results demonstrated participation in sports clubs during college is a key factor for school counselors’ physical activity. This is the first study to attempt to assess the impact of past participation in sports clubs at a college on current physical activity. Jacob et al. (2009) revealed that occupational physical activity was the largest contributor of adults’ weekly physical activity [33]. School counselors typically sit and interview or counsel students; thus, they spend more time sitting than classroom teachers. Our results demonstrated that 55.4% of the school counselors met the WHO criteria for physical activity. The data are slightly higher than Brito et al. 2012, which showed that 53.7% of public schoolteachers in the city of São Paulo, Brazil, met the physical activity level [22]. As indicated in Table 3, the moderate physical activity level group engaged in 526.6 ± 462.9 and 518.3 ± 529.5 MET-min/week of walking and vigorous activity, respectively, and the high physical activity level group, 2593.3 ± 2190.7 and 2540.0 ± 2509.2 MET-min/week, respectively. The low physical activity group spent the fewest MET-min/week on vigorous activity compared with other components of activity. These results suggest that counselors should be encouraged to engage in adequate vigorous activity per week to increase their physical activity levels.

Notably, the mere act of engaging in physical activity does not necessarily lead to sufficient cumulative physical activity. Because the IPAQ questionnaire calculates the accumulated physical activity over the past 7 days, it is not representative of physical activity from earlier periods. Therefore, designing a questionnaire to understand physical activity participation at different times is important. In this study, the results of Spearman correlation showed that the participation in sports clubs during college years was significantly associated with physical activity levels (rho = 0.234, *p* = 0.007, Table 4). In addition, school counselors with high levels of physical activity participated in sports clubs more often during college than those with low levels of physical activity (χ^2^ = 7.843, *p* = 0.02, Table 4) indicating that participation in sports clubs during college was a significant factor in the level of physical activity of school counselors.

Ragheb and Beard (1982) conducted a questionnaire-based study assessing individuals’ attitudes toward leisure activities [31] and determined that individuals’ cognitive, affective, and behavioral characteristics influence their attitudes toward leisure activities. Many studies have also explored the positive effects of positive attitudes toward leisure activities on activity participation [34,35,36]. In this study, we used the IPAQ to assess the physical activity levels of the school counselors in the seven days prior to survey administration to investigate the effects of physical activity participation, perceived barriers to physical activity, and attitudes toward physical activity on counselors’ physical activity levels. The factor analysis indicated that the component of physical activity participation was divided into active participation in sports clubs in college and participation in physical activities after employment (Table 1). Our results indicated that participation in sports clubs during college years was significantly associated with attitude toward physical activity (rho = 0.439, *p* = 0.000, Table 4). Moreover, as indicated in Table 3, participation in vigorous-intensity activities was common among the participants in the moderate and high physical activity level groups, which might be attributable to habits developed during college.

Self-determination theory (SDT) postulates that a person whose actions are autonomously motivated often exhibits superior mental health, well-being, and performance [37]. The theory of individual engagement in physical activity also supports SDT, with research suggesting that autonomous forms of motivation are positively associated with adaptive outcomes such as improved psychological well-being, greater behavioral persistence, and more objectively assessed behaviors or investments [38]. A study by Standage (2012) on school physical education and exercise reported that motivational processes predict activity levels and health-related well-being indices [39]. In addition, the effects of demographic variables, such as age and gender, on an individual’s participation in physical activity have often been studied [40,41,42]. Molanorouzi et al. (2015) argued that systematic differences between participation motives are related to demographic variables such as age and gender [43]. However, our results show that not only age (rho = −0.179, *p* = 0.042), but also the respondents’ participation in sports clubs during college years (rho = 0.234, *p* = 0.007), attitude toward physical activity (rho = 0.365, *p* = 0.000), body weight (rho = 0.221, *p* = 0.012), and self-perceived health status (rho = 0.186, *p* = 0.034) were significantly associated with their physical activity levels (Table 4).

In this study, we analyzed the effect of demographic variables on attitude toward physical activity and determined that self-perceived health status was significantly associated with attitude toward physical activity (Table 5). The school counselors perceived themselves as healthy were most likely to have an attitude toward physical activity (β = 0.168, t = 2.096, *p* = 0.038, VIF = 1.051, Table 5). Moreover, the participation in sports clubs during college was also an important factor associated with school counselors’ attitudes toward to physical activity (β = 0.409, t = 5.106, *p* = 0.000, VIF = 1.051, Table 5). Regression diagnostics indicated that all independent variables appear to contribute to predicting attitudes toward physical activity (Table 5). Collected, our results indicate that past sports club participation at college was an important factor in both current physical activity levels and attitudes towards physical activity. Although Curtis et al. (1999) have demonstrated that inter-school sport competition experience is a relatively strong predictor of sport participation later in life [44], the level of physical activity of those surveyed was not indicated. This is the first study to reveal that physical activity levels are associated with a history of participation in sports clubs as a college student. On the basis of our results, we strongly recommend that all students be encouraged to participate in sports clubs to maintain their physical activity habits in the future. Through participation in sports clubs, students not only increase their physical activity, but also learn to understand themselves, increase their self-confidence, and promote interpersonal relationships among peers, which will help them to develop social relationships after graduation.

Although the IPAQ was used to estimate the participants’ physical activity levels over the 7 days prior to survey administration, it is a self-administered questionnaire, and the researchers are unable to verify the responses and can only trust that the respondents answered each of the items honestly. Self-reported instruments are subject to bias, and future studies should use more objective measurements to reduce this bias. In addition, although this study included full-time school counselors employed in junior high schools in Kaohsiung city, the relatively small establishment of school counselors resulted in a limited sample size for this study. Future studies should survey the school counselors in several cities at the same time to obtain a larger sample size. Finally, school counselors were recruited from 2012; therefore, they are relatively young. Indeed, 80% of the school counselors in this study were ≤40 years of age, which may have influenced the relatively high rate of 55.4% of school counselors meeting the physical activity criteria. This is also subject to clarification when the sample size is expanded in the future. In addition to the sample size, future extensions to the analysis of school counselors’ physical activity could focus on the gap between participation in sports clubs, attitudes towards physical activity, and their physical activity.

## 5. Conclusions

In this study, our results demonstrated that the school counselors’ participation in sports clubs during college and the attitudes toward physical activity are important factors associated with physical activity level. School counselors with high levels of physical activity indicated greater participation in sports clubs during their time at college than those with low levels of physical activity. Moreover, the participation in sports clubs during college is an important factor in attitude toward physical activity. Hence, participation in sports clubs during college is a key factor for school counselors’ physical activity. Therefore, colleges or universities can develop school-based physical education-specific instruction and encourage students to establish and participate in sports clubs. A budget should also be allocated to build or maintain sports facilities and to support the operation of sports clubs. In this way, students will be able to learn good interpersonal skills and develop regular physical activity habits through participation in sports clubs.

## Figures and Tables

**Table 1 ijerph-19-05632-t001:** The questionnaire items of physical activity participation, perceived barriers, and attitude toward physical activity.

**Participation**	**Component ^@^**		
	**1**	**2**	
7	0.859		I participate in physical activities on weekdays.
8	0.796		I often engage in physical activities alone.
5	0.763		After getting a job as a school counselor, I regularly engage in physical activity.
6	0.73		I participate in physical activities on weekends.
2		0.922	In my college years (including graduate school years), I participated in sports clubs.
3		0.807	My current physical activity is related to the sports clubs in which I participated during my college years (including graduate school years).
1		0.786	During my college years (including graduate school years), I regularly participated in physical activities 2–3 times per week.
**Barrier**	**Component**		
	**1**		
1	0.819		I cannot participate in physical activities because I do not have a partner.
2	0.816		I cannot participate in physical activities because the venue is not well-equipped.
3	0.757		I cannot participate in physical activities because others do not support it.
**Attitude**	**Component**		
	**1**		
8	0.913		I care about my physical activity.
7	0.884		Through physical activity, I can find myself.
9	0.874		Participating in physical activities has given me valuable experience.
10	0.827		Physical activity makes me feel very comfortable and at ease.
6	0.74		I like the physical activities in which I choose to participate.
12	0.711		No matter how busy I am, I participate in physical activities.

Extraction method: Principal component analysis. ^@^ Rotation method: Varimax with Kaiser normalization.

**Table 2 ijerph-19-05632-t002:** Demographic characteristics of school counselors (n = 130).

Characteristic	n (M/F)	Percent (%)
Sex M F	26 104	20 80
Age (year) 21–30 31–40 ≥41	44 (11/33) 60 (14/46) 26 (1/25)	33.8 46.2 20.0
Height (cm) ≤150 151–160 161–170 ≥171	2 (0/2) 53 (0/53) 56 (8/48) 19 (18/1)	1.5 40.8 43.1 14.6
Body weight (kg) ≤50 51–60 61–70 ≥71	24 (0/24) 60 (2/58) 28 (10/18) 18 (14/4)	18.5 46.2 21.5 13.8
Marriage unmarried married	80 (16/64) 50 (11/39)	61.5 38.5
Education under graduated	55 (9/46) 75 (17/58)	42.3 57.7
Self-perceived health status unhealthy fair healthy	20 (3/17) 69 (7/62) 41 (16/25)	15.4 53.1 31.5

**Table 3 ijerph-19-05632-t003:** Physical activity level of school counselors.

Physical Activity Level (Score)	Activity Component	MET-min/Week (Mean ± SD)	n	Percent (%)
Low <600	Walking	127.4 ± 116.7	58	44.6
	Moderate	79.0 ± 113.2		
	Vigorous	54.5 ± 99.6		
	Cumulative *	260.9 ± 177.3		
Moderate ≤600~< 3000	Walking	526.6 ± 462.9	48	36.9
	Moderate	265.1 ± 343.3		
	Vigorous	518.3 ± 529.5		
	Cumulative *	1302.7 ± 603.8		
High ≥3000	Walking	2593.3 ± 2190.7	24	18.5
	Moderate	1776.7 ± 3166.7		
	Vigorous	2540.0 ± 2509.2		
	Cumulative *	6909.9 ± 3700.7		

Weekly walking MET-min score = 3.3 × frequency × activity time. Weekly moderate MET-min score = 4.0 × frequency × activity time. Weekly vigorous MET-min score = 8.0 × frequency × activity time. * Cumulative weekly physical activity score. SD: standard deviation.

**Table 4 ijerph-19-05632-t004:** Summary of correlation between physical activity levels and variables.

		MET-min	Participation in Sports Clubs During College	Attitude Toward Physical Activity
**Spearman’s rho** ** *p* ** **value**	MET-min		0.237	
			0.007 **^#^	
	Attitude toward physical activity	0.365	0.439	
		0.000 ***	0.000 ***	
	Age	−0.179	−0.341	−0.146
		0.042 *	0.000 ***^&^	0.098
	Body weight	0.221	−0.021	−0.079
		0.012 *	0.811	0.372
	Self-perceived health status	0.186 0.034 *	0.219 0.012 *^§^	0.257 0.003 **

*n* = 130; * *p* <0.05; ** *p* <0.01; *** *p* <0.001. ^#^ Kruskal–Wallis test: χ^2^ = 7.843, *p* = 0.02; high physical activity level > low physical activity level. ^&^ Kruskal–Wallis test: χ^2^ = 17.302, *p* = 0.000; 41 years > age 31–40 years. ^§^ Kruskal–Wallis test: χ^2^ = 10.434, *p* = 0.005; healthy > fair.

**Table 5 ijerph-19-05632-t005:** Multiple regression model results—determinants of school counselors’ attitudes toward physical activity.

Model	Unstandardized Coefficients		Standardized Coefficients	t	Sig.	Collinearity Statistics
	B	Std. Error	Beta			VIF
(Constant)	2.468	0.242		10.217	0.000	
Participation in sports clubs during college	0.275	0.054	0.409	5.106	0.000 ***	1.051
Self-perceived health status #	0.210	0.100	0.168	2.096	0.038 *	1.051

*n* = 130; * *p* <0.05; *** *p* <0.001, attitude toward physical activity = 2.468 + (0.275 * participation in sports clubs during college) + (0.210 * self-perceived health status). VIF: variance inflation factor. # post-hoc test: healthy > unhealthy; healthy > fair. Model summary: R^2^ = 0.0225, adjusted R^2^ = 0.213; analysis of variance: F = 18.476, *p* = 0.000. Durbin–Watson = 2.320; Cook’s distance = (0.000~0.108). Normality test (Shapiro-Wilk): Passed (*p* = 0.375). Constant variance test: Passed. (*p* = 0.301). Power of performed test with alpha = 0.050:1.000. All independent variables appear to contribute to predicting attitude toward physical activity (*p* < 0.05).

## Data Availability

The data are available upon request from the corresponding author.

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
