# Peer review of "Participation in Sports Clubs during College Is an Important Factor Associated with School Counselors’ Participation in Leisure Time Activities"

_ijerph, 2022, doi:10.3390/ijerph19095632_

Round 1

Reviewer 1 Report

GENERAL COMMENTS

INTRODUCTION

The introduction section should be improved by organizing the arguments. 

L43-69: Please try to link the sentences better, to improve the fluency of reading.

L56-69: Try to use a wider variety of connectors. Reduce the use of "Regarding".

L87-98: Try to better explain the justification of the present study.

METHODS

L-104: Please add reference.

RESULTS

Please, improve table’s format. It's difficult for the reader to interpret the results.

In this section, it can be observed in several situations, the redundancy of describing textually what the tables present. Please reconsider.

DISCUSSION 

I suggest you to improve the several considerations about the limitation of your study.  Moreover, add here which are the possible future lines.

CONCLUSION

How these findings can help population? Which specific practical implications can it have in college? 

Author Response

GENERAL COMMENTS

 Author response: We thank reviewer for your suggestions to make this manuscript better.

INTRODUCTION

The introduction section should be improved by organizing the arguments. 

L43-69: Please try to link the sentences better, to improve the fluency of reading.

 Author response: We have reorganized these sentences as suggested by reviewer.

L56-69: Try to use a wider variety of connectors. Reduce the use of "Regarding".

 Author response: We have replaced the regarding with other words as suggested by the reviewer.

L87-98: Try to better explain the justification of the present study.

 Author response: We have added explanations.

METHODS

L-104: Please add reference.

 Author response: We have added reference 25.

RESULTS

Please, improve table’s format. It's difficult for the reader to interpret the results.

Author response: We have rearranged table 3 and table 4.

In this section, it can be observed in several situations, the redundancy of describing textually what the tables present. Please reconsider.

 Author response: We have removed some descriptions from table 2 (section 3.1).

DISCUSSION 

I suggest you to improve the several considerations about the limitation of your study.  Moreover, add here which are the possible future lines.

  Author response: We have reorganized the paragraph as suggested by reviewer. Please find the last paragraph of the discussion section.

CONCLUSION

How these findings can help population? Which specific practical implications can it have in college? 

 Author response: We have revised it. Please find the highlighted sentence of the conclusion section.

Reviewer 2 Report

Overall, I believe this manuscript is now suitable for publication. However, I am recommending to the editors that the leading zeros be removed in all instances when reporting p-values because the value itself cannot be 1. 

Author Response

Overall, I believe this manuscript is now suitable for publication. However, I am recommending to the editors that the leading zeros be removed in all instances when reporting p-values because the value itself cannot be 1. 

Author response: We thank reviewer for positive comments. For numbers that are not greater than 1, we remove the zero before the decimal.

Reviewer 3 Report

Carefully reviewed your manuscript in terms of formatting, validation procedure, reliability issues, and confirmation of scientific research method in sport science. However, I do still feel that your manuscript needs much improvement in terms of those aspects listed. There are some of my comments regarding your study.

  1. Title statement – it cann’t improved regarding your focal variables instead of current one(complete sentence?)
  2. Your study does not confirm with submission guideline of IJRPH. You must thoroughly follow it prior to resubmission of your manuscript
  3. Your table should have proper title statement and note describing each component presented with your tables
  4. In table 2, where is the group of 51 years old or higher?
  5. I found multiple incomplete sentences throughout your manuscript. Your manuscript should be properly reviewed and edited by language polishing service in terms of wording and formatting

Author Response

Carefully reviewed your manuscript in terms of formatting, validation procedure, reliability issues, and confirmation of scientific research method in sport science. However, I do still feel that your manuscript needs much improvement in terms of those aspects listed. There are some of my comments regarding your study.

Author response: We thank reviewer for your suggestions to make this manuscript better.

  1. Title statement – it cann’t improved regarding your focal variables instead of current one(complete sentence?)

Author response: We have revised the title as suggested by another reviewer.

  1. Your study does not confirm with submission guideline of IJRPH. You must thoroughly follow it prior to resubmission of your manuscript

Author response: We have used the template format.

  1. Your table should have proper title statement and note describing each component presented with your tables

Author response: We have rearranged table 3 and table 4.

  1. In table 2, where is the group of 51 years old or higher?

Author response: We have made the correction and also removed some descriptions from table 2 (section 3.1).

  1. I found multiple incomplete sentences throughout your manuscript. Your manuscript should be properly reviewed and edited by language polishing service in terms of wording and formatting

Author response: In fact, we have asked English language editorial company to edit the text for readability and grammar before submission. If reviewer still feel that the revised manuscript needs to be re-edited, we will ask the English language editor again to edit the text.

Reviewer 4 Report

The article concerns the important issue of searching for factors that contribute to the shaping of pro-health behaviors of adults, especially those with sedentary work. The study was well planned and conducted using appropriate methods. The results show how important it is to promote physical activity among adolescents (students), because according to the proverb "as the twig is bent, so grows the tree" the habit of participating in physical activity developed in youth persists in adulthood.

The weakness of the study is that it was conducted only in one professional group. The analysis carried out only among school counsellors does not allow to answer the tree important question:

- whether the level of physical activity and individuals' attitudes toward leisure time physical activity in the school counsellors is similar or higher than in other professional groups with similarly high sedentary behavior at work;

- is the role of participation in sports clubs during college years or pedagogical education more important for the later pro-health behavior;

- is the positive role of participation in sport clubs during college years for future physical activity in free time specific only for school counsellors. 

It seems to me that in the title of the article, the phrase "post-employment physical activity" should be replaced with the phrase "leisure time physical activity".

Author Response

The article concerns the important issue of searching for factors that contribute to the shaping of pro-health behaviors of adults, especially those with sedentary work. The study was well planned and conducted using appropriate methods. The results show how important it is to promote physical activity among adolescents (students), because according to the proverb "as the twig is bent, so grows the tree" the habit of participating in physical activity developed in youth persists in adulthood.

The weakness of the study is that it was conducted only in one professional group. The analysis carried out only among school counsellors does not allow to answer the tree important question:

Author response: We thank reviewer for your suggestions to make this manuscript better.

- whether the level of physical activity and individuals' attitudes toward leisure time physical activity in the school counsellors is similar or higher than in other professional groups with similarly high sedentary behavior at work;

Author response: Thank you for your suggestions, which we have added in the first paragraph of the discussion section.

- is the role of participation in sports clubs during college years or pedagogical education more important for the later pro-health behavior;

Author response: Thank you for your suggestions, which we have added in the second last paragraph of the discussion section.

- is the positive role of participation in sport clubs during college years for future physical activity in free time specific only for school counsellors. 

Author response: Although our survey was conducted with school counsellors only, participation in sports clubs during college can be extended to all college students. We have added it to the last sentence of the conclusion section.

It seems to me that in the title of the article, the phrase "post-employment physical activity" should be replaced with the phrase "leisure time physical activity".

Author response: We have revised the title.

Round 2

Reviewer 3 Report

I don't see much improvement in terms of readability, formatting, gathering validity and reliability evidence. Moreover, what would be new (or so called scholarly contribution) to the domain of sports science and public health? I don’t find any good rationale from reading this article. Implication is “all students are encouraged to participate physical activity?” also your responsive statement to reviewers are not clearly describing what you have done throughout your manuscript. It is unfortunate but still not ready to be published in this journal.  

Author Response

Comments and Suggestions for Authors
- 3

I don't see much improvement in terms of readability, formatting, gathering validity and reliability evidence. Moreover, what would be new (or so called scholarly contribution) to the domain of sports science and public health? I don’t find any good rationale from reading this article. Implication is “all students are encouraged to participate physical activity?” also your responsive statement to reviewers are not clearly describing what you have done throughout your manuscript. It is unfortunate but still not ready to be published in this journal.  

Author response: We apologize for not being able to provide a detailed response to reviewers' comments. In the revised manuscript, the yellow highlight shows the previous revision and the blue highlight shows the current revision. Please refer to the following notes for this revision.

  1. In this revision, we have added a note on the shortcomings of classifying occupational groups when conducting physical activity surveys and a sentence at the end of the Introduction section to give the reader a more comprehensive understanding of the study when reading the introduction.
  2. In the Materials and Methods section, we have changed the order of sections 2.2 and 2.3 so that the presentation of methods and results is consistent.
  3. For the formatting of tables and manuscripts, the word template provided was used and the headings in Table 4 were revised. We believe that the journal will reconfirm the formatting if the manuscript is accepted.
  4. In the Discussion section, we added the benefits of students' participation in sports clubs.
  5. In the Conclusions, we also suggested practices for physical education and environment creation in colleges or universities.
  6. With regard to the fluency of English grammar, the journal agrees to provide English editing services upon acceptance of the manuscript.

This manuscript is a resubmission of an earlier submission. The following is a list of the peer review reports and author responses from that submission.

Round 1

Reviewer 1 Report

Overall, this was an interesting study. However, I have concerns regarding the development of the questionnaires, the validity of the multivariable regression and utility of the correlational analysis, and the use of language suggestive of causality among variables despite the cross-sectional study design. The manuscript must also improve the transparency of the methods by including far more details on questionnaire development and statistical analyses.

Abstract

  • You’ve stated that PA participation and attitude toward PA directly affects the participants’ PA levels – the use of the word ‘affected’ implies causation which cannot be determined using cross-sectional methods

Introduction

  • Do school counselors spend more time seated that other teachers at the school?
  • The use of the phrase “house-wives’ seems outdated
  • It is an interesting point that different occupations report varying levels of PA but as evidenced, teachers are not in the lowest PA bracket; which begs the question why focus on teachers specifically? It seems like an odd point to highlight given that your population of interest is teachers (and not labourers, who have the lowest rates of PA)

Methods

  • Was the questionnaire content based on previous literature on the participation and barriers of PA among teachers? How did you come up with these items?
  • I'm not sure I fully understand the purpose of the physical activity participation questions - they appear to be tapping into both past and current PA behaviours and I don't understand why these should be measured in a single score. It also seems redundant to correlate current PA behaviour to IPAQ scores are should be measuring the same thing.
  • More information about the specific models built is needed. In the multiple variable regressions, which variables were included? How did you determine which variables to enter into the model. The tables are not clear whether the different dimensions of the scales were both included at once, or if these are separate models.
  • Significance is set at one cut-off (not three) - but I believe you are just defining your symbols here which is not normally done within the methods text

Results

  • what was your justification for inviting only 156 counselors? Was there a sample size calculation performed? If not, please perform a power analysis.
  • The cronbach’s alpha for the barrier questionnaire is not great – please discuss and include a possible limitation to the validity of the results.
  • The header for 3.3, again, uses terminology suggestive of causation which should be modified (i.e. associated with PA levels rather than affect PA levels)
  • Were you actually examining the association with gender or did the questionnaire assess biological sex (i.e. was a binary option presented to respondents or were they able to self select from a variety of gender identities?)
  • Please keep your symbols consistent – in the methods you use asterisks to define different p-values but then use pound signs to define them in table 4
  • Table 5 - did you test for multilinearity? Any concern for overfitting? You have 11 parameters but only 130 participants - this is on the low end of the sample size requirements to support a model with this many parameters.
  • Table 8 was never mentioned in the results section nor was it described as an analysis in the methods section. This does to not appear to address your original research question and objectives.

Discussion

  • This first paragraph is repetitive - not necessary to repeat information from the introduction. This section should begin with a summary statement of the results found in your study.
  • The high rate of participants meeting WHO guidelines could also be signalling an issue with self-reported PA (social desirability bias almost always affects this variable) – this should be explained in your limitation section as well
  • There is a lot of repetitiveness throughout the discussion
  • I don't quite fully understand why “participation in exercise after employment” was specifically looked at separately from IPAQ scores in a correlational analysis - why would you be interested in exercise after employment if you have their habitual PA levels as reported in the IPAQ available?

Author Response

Overall, this was an interesting study. However, I have concerns regarding the development of the questionnaires, the validity of the multivariable regression and utility of the correlational analysis, and the use of language suggestive of causality among variables despite the cross-sectional study design. The manuscript must also improve the transparency of the methods by including far more details on questionnaire development and statistical analyses.

Author respond: We thank reviewer' suggestions, we have conducted a more detailed step-by-step analysis of the validity of the questionnaire and re-examined the reliability of the questionnaire. More care has also been taken in the use of words in the statistical results. Please refer to the highlighted section for the revised.

Abstract

  • You’ve stated that PA participation and attitude toward PA directly affects the participants’ PA levels – the use of the word ‘affected’ implies causation which cannot be determined using cross-sectional methods

Author respond: we have revised “affected” to “associated with”. (line 25, 26, 30)

Introduction

  • Do school counselors spend more time seated that other teachers at the school?

Author response: School counselors usually do not attend classes, but mainly interview or counsel students on a case-by-case basis, and the nature of their work is mainly seated.

  • The use of the phrase “house-wives’ seems outdated

Author response: we have revised it to “stay-at-home mothers. (line 86)

  • It is an interesting point that different occupations report varying levels of PA but as evidenced, teachers are not in the lowest PA bracket; which begs the question why focus on teachers specifically? It seems like an odd point to highlight given that your population of interest is teachers (and not labourers, who have the lowest rates of PA)

Author response: Indeed, the status of school counselors is that of teachers, not laborers. In Taiwan, full-time school counselors in junior high school are new faculty members since 2012. They are teachers, but their work is not mainly teaching, but is responsible for the second and third levels of prevention in the three-level counseling and prevention model of the school. Therefore, a relatively rare of physical activity studies were conducted on the physical activity of school counselors.

Methods

  • Was the questionnaire content based on previous literature on the participation and barriers of PA among teachers? How did you come up with these items?

Author response: The questionnaire on barriers and attitudes are based on literature with some modify, while the domain on participation is self-designed. (line 137-139)

Our second author is a school counsellor who had to attend regular supervision meetings where they shared their work and life experiences with each other. This is how the idea for the physical activity participation questionnaire was developed.

  • I'm not sure I fully understand the purpose of the physical activity participation questions - they appear to be tapping into both past and current PA behaviours and I don't understand why these should be measured in a single score. It also seems redundant to correlate current PA behaviour to IPAQ scores are should be measuring the same thing.

Author response: The International Physical Activity Questionnaire (IPAQ) represents the cumulative physical activity over the past seven days and is a quantitative representation of the results. The questionnaire was designed to understand the different stages of physical activity participation during college (including graduate school year), after graduation and after work. Behaviours that involve physical activity may not necessarily accumulate enough physical activity. However, a sufficient amount of physical activity means that there is physical activity participation. The most important thing we want to know is whether current physical activity is related to the development of exercise habits during the past school years. In this way, we can provide different strategies for physical activity for people at different stages of their lives. (Discussion section, paragraph 2)

  • More information about the specific models built is needed. In the multiple variable regressions, which variables were included? How did you determine which variables to enter into the model. The tables are not clear whether the different dimensions of the scales were both included at once, or if these are separate models.

Author response: We thank reviewers’ suggestion. We have incorporated the relevant variables into the regression model. The factors controlled for in the model are described in the revised manuscript. (Footnote in Table 4, 5, 6; Results section 3.3 and 3.4)

  • Significance is set at one cut-off (not three) - but I believe you are just defining your symbols here which is not normally done within the methods text

Author response: We thank reviewers’ suggestion. We have revised it. (line163, Materials and Methods section 2.4)

Results

  • what was your justification for inviting only 156 counselors? Was there a sample size calculation performed? If not, please perform a power analysis.

Author response: This is because there are only 156 school counselors in 89 public secondary schools. There were 137 school counselors who were willing to fill in the questionnaire and after deducting 7 invalid questionnaires, there were 130 valid questionnaires left. These 130 valid questionnaires already represent 83% of the school counselors. We have included the details of the factor analysis in the revised manuscript. (Materials and Methods section 2.1, 2.4)

  • The cronbach’s alpha for the barrier questionnaire is not great – please discuss and include a possible limitation to the validity of the results.

Author response: we have revised in the manuscript. (line 166-169; 391-395)

  • The header for 3.3, again, uses terminology suggestive of causation which should be modified (i.e. associated with PA levels rather than affect PA levels)

Author response: We thank reviewers’ suggestion. We have revised.

  • Were you actually examining the association with gender or did the questionnaire assess biological sex (i.e. was a binary option presented to respondents or were they able to self select from a variety of gender identities?)

Author response: We thank reviewers’ suggestion. In the questionnaire, we asked the gender according to the physical organ characteristics at birth. We have changed the wording.

  • Please keep your symbols consistent – in the methods you use asterisks to define different p-values but then use pound signs to define them in table 4

Author response: We have revised.

  • Table 5 - did you test for multilinearity? Any concern for overfitting? You have 11 parameters but only 130 participants - this is on the low end of the sample size requirements to support a model with this many parameters.

Author response: We have noticed the problem of sample size to variables ratio in the regression model. The proposed ratio of sample size to variables in the regression model is sample size: variables=10:1. The regression model in the manuscript follows the suggestion.

  • Table 8 was never mentioned in the results section nor was it described as an analysis in the methods section. This does to not appear to address your original research question and objectives.

Author response: Another reviewer thought that there were too many tables in the manuscript, so we removed the original Table 4 and presented the contents of Table 4 as Figure 1, and removed Table 6 and only descripted it in the revised manuscript.

Discussion

  • This first paragraph is repetitive - not necessary to repeat information from the introduction. This section should begin with a summary statement of the results found in your study.

Author response: We have revised the content of the discussion section.

  • The high rate of participants meeting WHO guidelines could also be signalling an issue with self-reported PA (social desirability bias almost always affects this variable) – this should be explained in your limitation section as well

Author response: We thank reviewers’ suggestion. We have added to the study limitations.

  • There is a lot of repetitiveness throughout the discussion

Author response: We have revised the content of the discussion section.

  • I don't quite fully understand why “participation in exercise after employment” was specifically looked at separately from IPAQ scores in a correlational analysis - why would you be interested in exercise after employment if you have their habitual PA levels as reported in the IPAQ available?

Author replay: As mentioned above, if it is known that maintaining physical activity is related to participation in sports clubs during the school years, then university students can be encouraged to participate in more sports clubs during their university years. The results could provide strategies to develop an exercise habit.

Reviewer 2 Report

Participation in sports clubs during college affects participation in exercise after employment in school counselors

Basic reporting

The study assessed the effect of partipation in sports clubs during college on the particpation in exercicie after employment. It is an interesting and well analysed topic. Whilst the study undoubtedly has merit, it is necessary to revise the style and the format. Moreover, I would recommend that a native co-author review the manuscript.

ABSTRACT

Please adapt the abstract according to the journal’s guide. 

INTRODUCTION

GENERAL COMENT: Despite a good review of the problem, I consider that it would be appropriate to enrich this section with the following topic: the decrease of physical fitness among students. This is a global problem and well-analysed topic for the scientific community. I suggest including some of the following references:

  • Ferrari, G. L. D. M., Matsudo, V. K. R., & Fisberg, M. (2015). Changes in physical fitness and nutritional status of schoolchildren in a period of 30 years (1980-2010). Revista Paulista de Pediatria, 33(4), 415–422. https://doi.org/10.1016/j.rppede.2015.03.001
  • Arboix-Alió, J., Buscà, B., Sebastiani, E. M., Aguilera-Castells, J., Marcaida, S., Garcia Eroles, L., & Sánchez López, M. J. (2020). Temporal trend of cardiorespiratory endurance in urban Catalan high school students over a 20 year period. PeerJ, 8, e10365. https://doi.org/10.7717/peerj.10365
  • Ortega, F. B., Artero, E. G., Ruiz, J. R., Espana-Romero, V., Jimenez-Pavon, D., Vicente-Rodriguez, G., Moreno, L. A., Manios, Y., Beghin, L., Ottevaere, C., Ciarapica, D., Sarri, K., Dietrich, S., Blair, S. N., Kersting, M., Molnar, D., Gonzalez-Gross, M., Gutierrez, A., Sjostrom, M., & Castillo, M. J. (2011). Physical fitness levels among European adolescents: the HELENA study. British Journal of Sports Medicine, 45(1), 20–29. https://doi.org/10.1136/bjsm.2009.062679
  • Tomkinson, G. R., Lang, J. J., & Tremblay, M. S. (2017). Temporal trends in the cardiorespiratory fitness of children and adolescents representing 19 high-income and upper middle-income countries between 1981 and 2014. British Journal of Sports Medicine, bjsports-2017-097982. https://doi.org/10.1136/bjsports-2017-097982

L39: Add reference

L-72-91: Divide it in two phrapgraph and provide the main purpose of the study.

MATERIAL AND METHODS

Divide this part in sections, according to the journal’s guide.

“Please, could you deeply present how you proceeding to select your sample? 
It is a representative sample? 

RESULTS

Please, improve table’s format and reduce the total number of tables.

I suggest removing Table 6 since its information can be provided in the text.

DISCUSSION

Please, start the discussion with the main purpose and the main findings.

In the discussion, the authors should be more specific to answer the study aim and address the main results. I find the idea of speculation interesting, but it is important not to shift the focus so far from the results found in the study.

L-247: Correct the reference

I suggest you to included several considerations about the limitation of your study. For example, there are several variables that you could include as a covariate? Is this a representative sample? Despite we can be able to understand what is happening among this population, how is our limitation when we do the inference of your results?”

CONCLUSIONS

The conclusion section should be further developed. Moreover a practical implications section should be included.

Author Response

Participation in sports clubs during college affects participation in exercise after employment in school counselors

Basic reporting

The study assessed the effect of partipation in sports clubs during college on the particpation in exercicie after employment. It is an interesting and well analysed topic. Whilst the study undoubtedly has merit, it is necessary to revise the style and the format. Moreover, I would recommend that a native co-author review the manuscript.

Author replay: We thank reviewer' suggestions. We have revised the manuscript based on reviewers' comments, especially the discussion section. Please refer to the highlighted section for the revised.

ABSTRACT

Please adapt the abstract according to the journal’s guide. 

Author response: we have revised. The heading in abstract has removed.

INTRODUCTION

GENERAL COMENT: Despite a good review of the problem, I consider that it would be appropriate to enrich this section with the following topic: the decrease of physical fitness among students. This is a global problem and well-analysed topic for the scientific community. I suggest including some of the following references:

  • Ferrari, G. L. D. M., Matsudo, V. K. R., & Fisberg, M. (2015). Changes in physical fitness and nutritional status of schoolchildren in a period of 30 years (1980-2010). Revista Paulista de Pediatria, 33(4), 415–422. https://doi.org/10.1016/j.rppede.2015.03.001
  • Arboix-Alió, J., Buscà, B., Sebastiani, E. M., Aguilera-Castells, J., Marcaida, S., Garcia Eroles, L., & Sánchez López, M. J. (2020). Temporal trend of cardiorespiratory endurance in urban Catalan high school students over a 20 year period. PeerJ, 8, e10365. https://doi.org/10.7717/peerj.10365
  • Ortega, F. B., Artero, E. G., Ruiz, J. R., Espana-Romero, V., Jimenez-Pavon, D., Vicente-Rodriguez, G., Moreno, L. A., Manios, Y., Beghin, L., Ottevaere, C., Ciarapica, D., Sarri, K., Dietrich, S., Blair, S. N., Kersting, M., Molnar, D., Gonzalez-Gross, M., Gutierrez, A., Sjostrom, M., & Castillo, M. J. (2011). Physical fitness levels among European adolescents: the HELENA study. British Journal of Sports Medicine, 45(1), 20–29. https://doi.org/10.1136/bjsm.2009.062679
  • Tomkinson, G. R., Lang, J. J., & Tremblay, M. S. (2017). Temporal trends in the cardiorespiratory fitness of children and adolescents representing 19 high-income and upper middle-income countries between 1981 and 2014. British Journal of Sports Medicine, bjsports-2017-097982. https://doi.org/10.1136/bjsports-2017-097982

Author response: We thanks reviewer provide the valuable references. We have added these references in the introduction section. (line 42-45; ref. 6-9)

L39: Add reference

Author response: We have added. (line 40, ref. 3)

L-72-91: Divide it in two phrapgraph and provide the main purpose of the study.

 Author response: We have revised. (line 91)

MATERIAL AND METHODS

Divide this part in sections, according to the journal’s guide.

Author response: We have revised. (Materials and Methods section 2.1 ~ 2.4)

“Please, could you deeply present how you proceeding to select your sample? 
It is a representative sample? 

Author response: Because there are only 156 school counselors in 89 public secondary schools. And there were 137 school counselors who were willing to fill in the questionnaire and after deducting 7 invalid questionnaires, there were 130 valid questionnaires left. These 130 valid questionnaires already represent 83% of the school counselors. (Materials and Methods section 2.1)

RESULTS

Please, improve table’s format and reduce the total number of tables.

Author response: We removed the original Table 4 and presented the contents of Table 4 as Figure 1, and removed Table 6 and only descripted it in the revised manuscript. The table’s format has improved in the revised manuscript.

I suggest removing Table 6 since its information can be provided in the text.

Author response: The table 6 was removed.

DISCUSSION

Please, start the discussion with the main purpose and the main findings.

Author response: We have revised the discussion section.

In the discussion, the authors should be more specific to answer the study aim and address the main results. I find the idea of speculation interesting, but it is important not to shift the focus so far from the results found in the study.

Author response: We have revised the discussion section.

L-247: Correct the reference

 Author response: This article was originally published in 1982.

I suggest you to included several considerations about the limitation of your study. For example, there are several variables that you could include as a covariate? Is this a representative sample? Despite we can be able to understand what is happening among this population, how is our limitation when we do the inference of your results?”

Author response: We thank reviewers’ suggestion. We understand reviewers’ concern. We have noticed the problem of sample size to variables ratio in the regression model. The proposed ratio of sample size to variables in the regression model is sample size: variables=10:1. Thus, the regression model in the manuscript follows the suggestions. We provide more information in the revised table note. We also have revised the limitation of the study in the manuscript. (line 387-401)

CONCLUSIONS

The conclusion section should be further developed. Moreover a practical implications section should be included.

Author response: We have revised the conclusions section.

Reviewer 3 Report

Dear authors,

The introduction section can be improved by organizing the arguments. I suggest that the reality, object of study, observed, is at the end of the topic.
I didn't observe the explanation of the objectives. The topic ends with the methods that were used.
The study involved human beings in the use of questionnaires, however I did not observe any mention of submission to a research ethics committee.
Table 1 should be an appendix to the study and its title, as in the others, is very summarized and insufficient about what is presented, which, in my opinion, is not clear enough.
The following Tables are unclear and poorly structured, as if they were drafts.
In the results, it is observed in several situations the redundancy of describing textually what the tables present.
Line 181: First uses the expression "almost equal" in the MET parameter comparison. A statistical test could be used for this purpose and not in this subjective way.
Lines 193 to 195: the indication in Table 6 does not correspond to its contents. I think it refers to the contents of Table 7.
Table 5, last column: what is VIF?
Tables 7 and 8, last columns: the post-hoc test suddenly appears.
The sharpness of Figure 1 is seriously compromised.
In the topic referring to the Discussion, there is a substantial return to the introduction, repeating what was said. The paragraphs are extremely long, a characteristic that makes understanding difficult.

Author Response

Dear authors,

The introduction section can be improved by organizing the arguments. I suggest that the reality, object of study, observed, is at the end of the topic.

Author response: We thank reviewers’ suggestion. We have revised. Please refer to the highlighted section.

I didn't observe the explanation of the objectives. The topic ends with the methods that were used.

Author response: We have revised. (line 93-102)

The study involved human beings in the use of questionnaires, however I did not observe any mention of submission to a research ethics committee.

Author response: The Human Research Ethics Committee of National Cheng Kung University, Taiwan was consulted by telephone. Because the questionnaire is a non-named, non-interactive, non-interventional study, and no specific individuals can be identified from the information in the questionnaire, it is exempt from being sent to the Human Research Ethics Committee. Therefore, it was not sent to the Human Research Ethics Committee.

Table 1 should be an appendix to the study and its title, as in the others, is very summarized and insufficient about what is presented, which, in my opinion, is not clear enough.

Author respond: We have improved.

The following Tables are unclear and poorly structured, as if they were drafts.

Author response: Another reviewer thought that there were too many tables in the article, so we removed the original Table 4 and presented the contents of Table 4 as Figure 1, and removed Table 6 and only descripted it in the revised manuscript. The format of the table has been improved.

In the results, it is observed in several situations the redundancy of describing textually what the tables present.

Author response: The idea is that we can read the results from the text without using tables. This is why we have given some descriptions.

Line 181: First uses the expression "almost equal" in the MET parameter comparison. A statistical test could be used for this purpose and not in this subjective way.

Author response: We have revised. Please refer to the highlighted in results section 3.2.

Lines 193 to 195: the indication in Table 6 does not correspond to its contents. I think it refers to the contents of Table 7.

Author respond: The other reviewer suggests to remove Table 6. Thus, the Table 6 was removed. The information was provided in the text. Please refer to the highlighted in results section 3.3 and 3.4.

Table 5, last column: what is VIF?

Author response: A footnote of VIF has been added.

Tables 7 and 8, last columns: the post-hoc test suddenly appears.

Author response: The post-hoc test is described in the Materials and Methods section. (line 155)

The sharpness of Figure 1 is seriously compromised.

Author response: We have improved the clarity of Figure 1.

In the topic referring to the Discussion, there is a substantial return to the introduction, repeating what was said. The paragraphs are extremely long, a characteristic that makes understanding difficult.

Author response: We have revised the discussion section.

Round 2

Reviewer 1 Report

I appreciate the authors attempts to address my original concerns. However, there remain gaps in the reporting and concerns regarding the analyses and strength of conclusions. 

  • Title: “influences” is a synonym for “affects” – change to “is associated with”
  • I am confused with your argument in your response document regarding the physical activity participation questionnaire. I understand the purpose of component 2 of the questionnaire. However, as you state, PA participation is not equivalent to sufficient PA so it is not clear why you would be interested in ‘current PA participation’ as constructed in component 1 of your participation questionnaire when the IPAQ gives you this information AND tells you whether their current PA is sufficient. Therefore, I still question the utility of component 1 of the PA participation questionnaire relative to the IPAQ.
  • Should include a power analysis since a sample size calculation was not conducted.
  • Even if you are following guideline of 10:1 participants to variables, you should still follow appropriate model testing and assumption testing including an evaluation of co- and multilinearity – this is standard procedure for regression analyses. You have reported VIFs but have not commented on their meaning – please include in your methods and a brief statement in your results
  • Given both the face validity and the results of your construct validity for your participation questionnaire it appears that it is tapping into 2 dimensions so it should not be entered into the regression as a single score.
  • Your statement on page 10 that the IPAQ does not represent ongoing PA is incorrect as the IPAQ has been designed and validated for that exact purpose of measuring daily PA
  • I’m still not clear on the relevancy of Table 6 to your research objectives and it was not mentioned in your statistical analysis section so it appears entirely post-hoc (suggest removing it altogether)
  • Again, the information gained from the correlation analysis is null since your PA participation questionnaire is not validated and the IPAQ is validated (and is more likely to accurately represent their PA habits which is what you appear to be interested in helping school counselors improve/maintain). 
  • Your conclusion is still implying causation i.e. the use of the word “influencing”

Author Response

Point-by-point response to reviewer 1:

I appreciate the authors attempts to address my original concerns. However, there remain gaps in the reporting and concerns regarding the analyses and strength of conclusions. 

Response: Thank you for your overall comments and for the valuable time you spent on our manuscript. We have revised your comments and made corrections accordingly, as shown below. The changes are highlighted within the manuscript.

  • Title: “influences” is a synonym for “affects” – change to “is associated with”

Response: We have revised the title of the revised manuscript. " Participation in sports clubs during college is an important factor associated with participation in physical activity after employment in school counselors”

  • I am confused with your argument in your response document regarding the physical activity participation questionnaire. I understand the purpose of component 2 of the questionnaire. However, as you state, PA participation is not equivalent to sufficient PA so it is not clear why you would be interested in ‘current PA participation’ as constructed in component 1 of your participation questionnaire when the IPAQ gives you this information AND tells you whether their current PA is sufficient. Therefore, I still question the utility of component 1 of the PA participation questionnaire relative to the IPAQ.

Response: Thank you for reminding us of this issue. We understand reviewer’ concerns, we have revised all the participation in the manuscript to component 2 (participation in sports clubs during college) scores. In addition, questions with two components in the attitude questionnaire have been revised to remove three items (component 2), making it a single component. We also revised Table 5 by revising the regression model for the factors associated with predicting attitudes toward physical activity. In the revised manuscript, we conducted new regression model and revised the presentation of Table 4 and 5.

  • Should include a power analysis since a sample size calculation was not conducted.

Response: We have added a sample size calculation to the Materials and Methods section. (2.1 and 2.4, highlighted).

  • Even if you are following guideline of 10:1 participants to variables, you should still follow appropriate model testing and assumption testing including an evaluation of co- and multilinearity – this is standard procedure for regression analyses. You have reported VIFs but have not commented on their meaning – please include in your methods and a brief statement in your results

Response: We have followed the suggestions of reviewer and re-screened the model by stepwise regression method and tested the validity of the model by power analysis. The presentation of Table 4 and Table 5 has also been revised. Please find section 2.4, 3.3 and 3.4, highlighted.

  • Given both the face validity and the results of your construct validity for your participation questionnaire it appears that it is tapping into 2 dimensions so it should not be entered into the regression as a single score.

Response: We have revised this concern. In the revised manuscript, we removed the scores from the current participation component of the questionnaire. The score of the participation in sports clubs during college was collected from item 1, 2, and 3 (component 2).

  • Your statement on page 10 that the IPAQ does not represent ongoing PA is incorrect as the IPAQ has been designed and validated for that exact purpose of measuring daily PA

Response: We agree with the reviewer that the meaning of this sentence has been revised. “Because the IPAQ questionnaire calculates the accumulated physical activity over the past 7 days, it is not representative of physical activity from earlier periods.” Please find the second paragraph in discussion section, highlighted.

  • I’m still not clear on the relevancy of Table 6 to your research objectives and it was not mentioned in your statistical analysis section so it appears entirely post-hoc (suggest removing it altogether)

Response: We have removed the Table 6.

  • Again, the information gained from the correlation analysis is null since your PA participation questionnaire is not validated and the IPAQ is validated (and is more likely to accurately represent their PA habits which is what you appear to be interested in helping school counselors improve/maintain). 

Response: We have revised the score of the questionnaire. The scores for the participation part are only calculated from component 2 (participation in sports clubs during college). The scores of the Attitude Questionnaire have also been revised. We have revised the regression model in Table 5 to predict the factors associated with attitudes toward physical activity.

  • Your conclusion is still implying causation i.e. the use of the word “influencing”

Response: We have revised the conclusions section.

Reviewer 2 Report

The authors were concerned with improving the work through suggestions, which greatly increased the quality of the article. However, there are some issues that should be improved to increase the manuscript quality.

METHODS

Please, introduce the Sample section (2.1 Sample).

I understand your explanation about the number of counselours, but please, introduce the Sample size calculation. I suggest using G-Power software or similar.

DISCUSSION

Please again, start the discussion with the main purpose and the main findings of your investigation.

CONCLUSIONS

Again, the conclusion section should be further developed.

You should include a practical implications section.

Author Response

Point-by-point response to reviewer 2:

The authors were concerned with improving the work through suggestions, which greatly increased the quality of the article. However, there are some issues that should be improved to increase the manuscript quality.

Response: Thank you for your overall comments and for the valuable time you spent on our manuscript. We have revised your comments and made corrections accordingly, as shown below. The changes are highlighted within the manuscript.

METHODS

Please, introduce the Sample section (2.1 Sample).

I understand your explanation about the number of counselours, but please, introduce the Sample size calculation. I suggest using G-Power software or similar.

 Response: We appreciate the information about G*Power Software. We have added a sample size calculation to the Materials and Methods section. (2.1 and 2.4, highlighted).

DISCUSSION

Please again, start the discussion with the main purpose and the main findings of your investigation.

 Response: We have revised the beginning of the discussion section.

CONCLUSIONS

Again, the conclusion section should be further developed.

You should include a practical implications section.

 Response: We have revised the conclusions section.

Reviewer 3 Report

I was able to verify that the suggestions and considerations presented were almost entirely accepted.
However, I believe that the issue of not needing submission to the research ethics committee should be addressed. 

Author Response

Point-by-point response to reviewer 3:

I was able to verify that the suggestions and considerations presented were almost entirely accepted.
However, I believe that the issue of not needing submission to the research ethics committee should be addressed. 

Response: Thank you for your kind overall evaluation and your valuable time spent on our manuscript. We have adopted the editor's recommendation to report the justification for not submitting to an ethics committee in the material and methods section.